# Properties and Functions of Fibroblasts and Myofibroblasts in Myocardial Infarction

**DOI:** 10.3390/cells11091386

**Published:** 2022-04-20

**Authors:** Harikrishnan Venugopal, Anis Hanna, Claudio Humeres, Nikolaos G. Frangogiannis

**Affiliations:** The Wilf Family Cardiovascular Research Institute, Albert Einstein College of Medicine, 1300 Morris Park Avenue Forchheimer G46B, Bronx, NY 10461, USA; harikrishnan.venugopal@einsteinmed.edu (H.V.); anis.hanna@einsteinmed.org (A.H.); claudio.humeres@einsteinmed.org (C.H.)

**Keywords:** fibroblast, myofibroblast, myocardial infarction, cytokine, fibrosis, angiogenesis, remodeling, extracellular matrix

## Abstract

The adult mammalian heart contains abundant interstitial and perivascular fibroblasts that expand following injury and play a reparative role but also contribute to maladaptive fibrotic remodeling. Following myocardial infarction, cardiac fibroblasts undergo dynamic phenotypic transitions, contributing to the regulation of inflammatory, reparative, and angiogenic responses. This review manuscript discusses the mechanisms of regulation, roles and fate of fibroblasts in the infarcted heart. During the inflammatory phase of infarct healing, the release of alarmins by necrotic cells promotes a pro-inflammatory and matrix-degrading fibroblast phenotype that may contribute to leukocyte recruitment. The clearance of dead cells and matrix debris from the infarct stimulates anti-inflammatory pathways and activates transforming growth factor (TGF)-β cascades, resulting in the conversion of fibroblasts to α-smooth muscle actin (α-SMA)-expressing myofibroblasts. Activated myofibroblasts secrete large amounts of matrix proteins and form a collagen-based scar that protects the infarcted ventricle from catastrophic complications, such as cardiac rupture. Moreover, infarct fibroblasts may also contribute to cardiac repair by stimulating angiogenesis. During scar maturation, fibroblasts disassemble α-SMA+ stress fibers and convert to specialized cells that may serve in scar maintenance. The prolonged activation of fibroblasts and myofibroblasts in the infarct border zone and in the remote remodeling myocardium may contribute to adverse remodeling and to the pathogenesis of heart failure. In addition to their phenotypic plasticity, fibroblasts exhibit remarkable heterogeneity. Subsets with distinct phenotypic profiles may be responsible for the wide range of functions of fibroblast populations in infarcted and remodeling hearts.

## 1. Introduction

Myocardial infarction typically results from the acute thrombotic occlusion of a coronary artery due to plaque rupture or erosion and is a major cause of morbidity and mortality worldwide. The loss of perfusion in the territory subserved by the occluded vessel leads to massive cardiomyocyte necrosis, overwhelming the negligible regenerative capacity of the adult human heart [1]. Thus, repair of the infarcted myocardium requires the mobilization and activation of reparative fibroblasts and myofibroblasts that deposit extracellular matrix proteins, preventing the catastrophic rupture of the ventricle [2,3,4]. Fibroblast activity in the infarct needs to be tightly regulated in order to prevent unrestrained matrix deposition in the border zone and in the remote remodeling myocardium [5]. Thus, perturbations in the inhibitory signals that mediate fibroblast deactivation after scar formation may contribute to the expansion of fibrosis and may cause adverse remodeling and heart failure. Understanding the molecular signals regulating the phenotypic transitions of fibroblasts in healing infarcts is a major priority in cardiovascular research and has important therapeutic implications for patients with myocardial infarction or heart failure. This review manuscript discusses the patterns of activation, mechanisms of phenotypic modulation, and functional role of fibroblasts in the infarcted myocardium. Moreover, we attempt to identify fibroblast-related therapeutic targets in myocardial infarction, acknowledging the major challenges in the implementation of anti-fibrotic interventions in cardiac repair.

## 2. The Fibroblasts in Normal Adult Hearts

The identification of fibroblasts is often credited to Virchow, who, in the mid-19th century, described a population of spindle-shaped connective tissue cells that synthesize collagen. More than 150 years later, the definition of the fibroblast remains morphological and functional in nature. Fibroblasts are typically defined as cells of mesenchymal origin that reside in connective tissues and serve primarily to maintain the ECM network [6]. However, recent single-cell transcriptomic studies contributed to the appreciation that fibroblast populations exhibit remarkable intra- and inter-organ heterogeneity [7,8,9]. Differences in the ECM gene expression profiles between fibroblasts from different organs suggest that fibroblasts may tailor their ECM production in an organ-specific manner [8].

The notion that the mammalian heart contains abundant fibroblasts was first supported by morphological and ultrastructural studies. Early investigations suggested that fibroblasts may be the predominant non-cardiomyocyte cell type found in normal mammalian myocardium and may even outnumber cardiomyocytes [10,11]. Moreover, a flow cytometric study showed that in adult mouse hearts 27% of myocardial cells can be identified as fibroblasts based on the expression of DDR2, whereas only 7% of the cells are CD31+ endothelial cells [12]. In contrast to these findings, a more recent investigation using fibroblast reporter mouse models and cell-specific antibodies showed that <20% of myocardial non-cardiomyocytes can be identified as fibroblasts and are greatly outnumbered by endothelial cells, which represent >60% of non-cardiomyocytes [13]. The predominance of endothelial cells in the non-cardiomyocyte fraction of myocardial cells is consistent with the very high vascularity of the heart, which is suggested by a vast amount of histopathological data in several different species. It should be emphasized that conflicting data of the relative abundance of cardiac fibroblasts reported in various studies may be explained by the use of different strategies for the identification of myocardial cells or by age-, sex-, and species-specific differences. The reliable and consistent identification of fibroblasts is hampered by the absence of sensitive and specific markers for fibroblasts that do not exhibit reactivity with other cell types. Commonly used markers have significant limitations. Vimentin labels all fibroblasts [14] but is non-specific and is expressed by all cells of mesenchymal origin [15,16]. On the other hand, α-smooth muscle actin (α-SMA) and periostin are markers for myofibroblasts [17,18] and do not label fibroblasts in the absence of myofibroblast conversion. α-SMA lacks specificity for the fibroblast lineage, as it is also highly expressed in vascular smooth muscle cells. Fibroblast-specific protein (FSP)-1, considered specific for fibroblasts in some studies, is also highly expressed by several other cell types, including macrophages, T cells, and activated vascular cells [19,20]. The most reliable markers for fibroblasts in the myocardium are the transcription factor Tcf21 [9,21] and the growth factor receptor platelet-derived growth factor receptor (PDGFR)α [22]. In mice, the PDGFRα^EGFP^ reporter line was found to specifically label interstitial and perivascular fibroblasts without any overlap with vascular mural cells [23]. The remarkable phenotypic plasticity exhibited by cardiac fibroblasts under conditions of stress adds additional challenges to the identification of fibroblasts using a single marker. Thus, the reliable characterization of cardiac fibroblasts may require the combined use of fibroblast-related markers along with exclusion criteria, confirming the lack of expression of hematopoietic and vascular cell markers.

## 3. Homeostatic Functions of Fibroblasts in the Adult Mammalian Heart

Although fibroblasts have been implicated in cardiac development, by regulating cardiomyocyte proliferation [24] and contributing to the formation of the extracellular matrix (ECM) network [21], their role in the homeostatic function of the adult myocardium remains enigmatic. Considering the prominent role of cardiac fibroblasts in matrix deposition and metabolism, their role in the maintenance of the cardiac ECM network is intuitive and may involve both matrix synthesis and degradation. Although systematic investigations on the role of cardiac fibroblasts in preserving the structure and function of the adult mammalian heart have not been performed, two lines of evidence support their involvement in the maintenance of the myocardial ECM network. First, fibroblast-specific loss of PDGFRα was associated with a 50% reduction in the number of cardiac fibroblasts and resulted in perturbations in the basement membrane without affecting systolic ventricular function [25]. Second, a fibroblast-specific loss of Smad3 modestly but significantly reduced myocardial collagen content without affecting ventricular function, geometry, or heart rate [26]. The absence of functional defects in both these studies may reflect the partial perturbation of fibroblast actions, and the limited duration of follow-up may not be sufficient for the assessment of the effects of fibroblasts on ventricular function. 

In addition to their effects on matrix synthesis and metabolism, cardiac fibroblasts may also transduce key homeostatic signals to other cell types, including cardiomyocytes and vascular cells [27,28], and may participate in the conduction of the electrical impulse. Fibroblasts express high levels of connexins and may contribute to the electrical coupling of cardiomyocytes from different layers, thus facilitating the synchronization of contraction [29]. Although this concept is supported by experiments in complex heterocellular co-culture models [30], the potential in vivo role of fibroblasts in synchronizing the ventricular contractile response has not been directly tested. 

The maintenance of fibroblast quiescence under homeostatic conditions also appears to be an active process that involves the baseline stimulation of inhibitory signals. For example, the Hippo pathway kinases large tumor suppressor kinase (LATS)1 and LATS2 were found to restrain fibroblast activation in normal adult hearts, protecting them from spontaneous fibrosis [31].

## 4. The Reparative Response after Myocardial Infarction: From Inflammation to Fibrosis

The reparative response following myocardial infarction can be divided into three overlapping phases: the inflammatory phase, the proliferative phase, and the maturation phase. Repair of the infarcted heart ultimately results in the replacement of dead myocardium with a mature collagen-based scar that lacks contractile capacity but protects the ventricle from catastrophic complications, such as cardiac rupture. As the infarct heals, the ventricle remodels, exhibiting dilation and hypertrophy of non-infarcted segments. The severity of post-infarction remodeling is dependent not only on the size of the infarct (and the amount of myocardium that is lost) but also on the qualitative characteristics of the reparative response. Perturbed repair may result in the formation of a scar with a low tensile strength, thus promoting chamber dilation and contributing to the pathogenesis of systolic heart failure. On the other hand, excessive fibrogenic activation may result in accentuated collagen deposition in the infarct border zone, increasing diastolic dysfunction. 

Fibroblasts are the main producers of collagens in the infarcted heart [32]. Fibroblasts undergo dramatic phenotypic transitions during the three phases of infarct healing [33] (Figure 1). During the inflammatory phase, resident cardiac fibroblasts in the ischemic region acquire a pro-inflammatory phenotype and may mediate the recruitment of leukocytes through the secretion of cytokines [34]. Moreover, cytokine stimulation induces the expression of matrix-degrading proteases in cardiac fibroblasts, contributing to the degradation of the matrix network [35]. During the proliferative phase, the fibroblast population in the infarct expands through the proliferation and migration of resident fibroblasts from the non-infarcted segments. A large fraction of infarct fibroblasts convert to myofibroblasts, expressing contractile proteins, such as α-smooth muscle actin (α-SMA), and synthesize large amounts of ECM proteins [36,37]. Finally, during the maturation phase, myofibroblasts disassemble their α-SMA-containing stress fibers and convert into specialized cells that serve to maintain the scar and have been termed matrifibrocytes [38]. 

## 5. Fibroblasts during the Inflammatory Phase: Pro-Inflammatory and Matrix-Degrading Functions

Cardiomyocyte necrosis releases damage-associated molecular patterns (DAMPs) and activates innate immune pathways, resulting in the induction and release of pro-inflammatory cytokines and chemokines and the subsequent recruitment of neutrophils and monocytes. In vitro, cardiac fibroblasts are known to respond to toll-like receptor (TLR) activation and to interleukin (IL)-1 stimulation by producing and secreting large amounts of pro-inflammatory cytokines [39,40,41,42,43]. However, considering that several other myocardial cell types, including resident macrophages [44], mast cells [45], endothelial cells [46], and cardiomyocytes [47], have been demonstrated to secrete large amounts of pro-inflammatory mediators in the infarcted heart, the relative contribution of fibroblasts in the activation of the post-infarction inflammatory response in vivo is unclear. Myocardial ischemia activates the NLR family pyrin domain containing (NRLP)3 inflammasome in cardiac fibroblasts [48,49], which may serve as an important source of active IL-1β. Moreover, the multifunctional pro-inflammatory granulocyte/macrophage colony-stimulating factor (GM-CSF) has been identified as a major fibroblast-derived signal that promotes the infiltration of the infarct with pro-inflammatory leukocytes [50]. Although infarct fibroblasts likely contribute to post-infarction inflammation by producing a wide range of pro-inflammatory mediators, cell-specific interventions examining the role of fibroblast-derived cytokines in leukocyte recruitment in the infarcted heart have not been performed. Thus, to what extent cardiac fibroblasts also contribute to the marked upregulation of chemokines and cytokines in the infarct environment remains unknown. 

In addition to producing cytokines and chemokines, fibroblasts have been also suggested to contribute to the degradation of the damaged matrix in the infarcted heart by producing proteases (Figure 2). Hypoxic conditions and stimulation with pro-inflammatory cytokines, such as IL-1 and tumor necrosis factor (TNF)-α, induce matrix metalloproteinase (MMP) synthesis and release by cardiac fibroblasts [35,51,52]. The early matrix-degrading actions of fibroblasts during the inflammatory phase of infarct healing may promote the clearance of damaged matrix proteins from the infarct, setting the stage for the later activation of a reparative program and the formation of a collagen-based scar. Moreover, the induction of membrane-bound metalloproteinases on the surfaces of infarct fibroblasts may promote a migratory phenotype [53], facilitating the mobilization of fibroblasts to repair the infarct.

### Do Fibroblasts in the Infarct Zone Die?

Although the time course of cardiomyocyte death in the infarcted myocardium has been extensively investigated [54], the susceptibility of other resident myocardial cell types to ischemic death remains poorly understood. The more susceptible subendocardial cardiomyocytes exhibit irreversible changes as early as 20–30 min after coronary occlusion; these changes culminate in death [55]. More prolonged ischemic intervals stimulate a “wavefront” of cardiomyocyte death that ultimately results in transmural infarction 5–6 h after coronary occlusion [56]. It has been suggested that cardiac fibroblasts express much higher levels of anti-apoptotic B cell lymphoma (Bcl)-2 than fibroblasts from other sites and may be highly resistant to death under conditions of stress [57]. However, the patterns and mechanisms of cell death in infarct fibroblasts have not been systematically studied. Lineage-tracing studies showed that in a mouse model of non-reperfused infarction, Tcf21 lineage cells were markedly reduced in the infarct zone as early as 2 days after coronary ischemia [38]. Apoptotic fibroblasts were identified in the infarct zone at this timepoint, suggesting that at least a fraction of the resident fibroblasts may die after prolonged ischemia and that the expansion of the fibroblast population during the proliferative phase may reflect the recruitment of cells from other regions [38]. To what extent reperfusion salvages the resident cardiac fibroblast population and whether such an effect plays a role in repair has not been investigated. It has also been suggested that infarct fibroblasts may protect cardiomyocytes from ischemic injury. However, this notion is based on associative evidence, and the mechanisms for such protective effects remain unknown [58]. 

## 6. Fibroblasts during the Proliferative Phase of Infarct Healing: Myofibroblast Conversion and Acquisition of a Migratory, Proliferative, and Matrix-Synthetic Phenotype

### 6.1. The Potential Role of Infarct Fibroblasts in Phagocytosis and Suppression of Inflammation

The transition from the inflammatory to the proliferative phase requires the induction and release of anti-inflammatory signals. Phagocytosis of dead cells by macrophages is a central event in the suppression and resolution of inflammation, resulting in the release of anti-inflammatory mediators, such as IL-10 and transforming growth factor (TGF)-β [59]. The cytokine-mediated activation of an anti-inflammatory phenotype in macrophages [60] and the recruitment of T cells with anti-inflammatory properties (such as regulatory T cells/Tregs) [61,62] also contribute to the negative regulation of post-infarction inflammation. Whether fibroblasts secrete anti-inflammatory mediators, contributing to the transition from the inflammatory phase to the proliferative phase, remains unknown. Some experimental evidence suggests that activated fibroblasts may serve as phagocytes, engulfing apoptotic cells from the infarct zone [63]; this process could potentially trigger the release of anti-inflammatory mediators. However, considering the prominent role of professional phagocytosis in this process, the relative role of fibroblasts as phagocytic and pro-resolving cells remains unknown. 

### 6.2. The Cellular Origins of Fibroblast Expansion in the Infarcted Myocardium

The transition to the proliferative phase of cardiac repair is characterized by a marked increase in the number of fibroblasts in the infarct and the border zone and by the emergence of activated myofibroblasts, fibroblast-like cells that express contractile proteins (such as α-SMA) and deposit large amounts of ECM [36,37,64]. It has been suggested that, in addition to the activation of a proliferative program in fibroblasts residing in the border zone of the infarct and to the migration of fibroblasts from non-ischemic myocardium, several other cell types may convert to fibroblasts, contributing to the reparative response. Thus, several studies have suggested that endothelial cells [65] or myeloid cells (including hematopoietic myeloid progenitors or macrophages) [66,67] may contribute to fibroblast expansion in the infarcted heart by converting to ECM-producing fibroblasts. However, in recent years, several studies using robust lineage-tracing experiments showed that the majority of infarct fibroblasts and myofibroblasts originate from resident fibroblast-like cells, with minor contributions from hematopoietic or endothelial cells [68,69]. 

### 6.3. Myofibroblast Conversion

Myofibroblast conversion is a critical cellular event in the reparative fibrosis of the infarcted heart. In animal models of myocardial infarction, myofibroblast transdifferentiation is first noted during the proliferative phase (Figure 3) and is predominantly localized in the infarct border zone, where activated myofibroblasts form well-organized arrays [70]. The absence of myofibroblasts during the inflammatory phase of cardiac repair has been attributed to the inhibitory effects of pro-inflammatory cytokines, such as IL-1β, on fibroblast α-SMA synthesis [35]. The earliest marker of myofibroblast conversion in infarct fibroblasts is the expression of the matricellular protein periostin, which is followed by increased synthesis of contractile proteins, such as α-SMA and the embryonic isoform of smooth muscle myosin heavy chain (SMemb) [36,71]. Myofibroblast conversion is associated with the acquisition of a matrix synthetic phenotype, characterized by high levels of expression of both structural collagens and matricellular proteins. The potential link between the expression of contractile proteins and the activation of a matrix-producing program in activated myofibroblasts is unclear.

Healing the infarct requires a highly plastic interstitium in which fibroblasts, macrophages, and vascular cells can migrate, thus forming a scar. The migration of fibroblasts in the infarcted heart involves actions of cytokines and growth factors [40,72,73,74] that stimulate the continuous formation and disassembly of adhesions between the ECM and fibroblast surface proteins. Several members of the integrin family are implicated in fibroblast migration in the infarct, interacting with specialized matrix proteins that are secreted in the infarcted myocardium [75,76]. Moreover, migratory fibroblasts express and activate membrane-bound proteases that serve to create conduits within the ECM network [53,77]. 

### 6.4. Molecular Signals Serving as Activators of Infarct Myofibroblasts

A wide range of cytokines, growth factors, neurohumoral mediators, and matricellular proteins have been implicated in the activation of infarct fibroblasts and myofibroblasts [78,79]. Although there is a vast amount of in vitro evidence supporting the role of each one of these mediators in the activation of infarct myofibroblasts, the in vivo significance of these activating signals is poorly understood, as cell-specific in vivo interventions are lacking.

#### 6.4.1. Cytokines

The pro-inflammatory cytokines IL-1, TNF-α, and IL-6 stimulate cytokine and chemokine synthesis by cardiac fibroblasts, promoting a matrix-degrading phenotype (Table 1). Although, in the short term, these effects delay myofibroblast conversion and prevent premature matrix deposition until the infarct is cleared of dead cells, the long-term actions of the pro-inflammatory cytokines are fibrogenic through several different mechanisms. First, stimulation with pro-inflammatory cytokines may promote the induction of fibrogenic growth factors, such as TGF-β [80]. Second, the cytokine-mediated induction and activation of proteases may contribute to the activation of latent TGF-β stores [81]. Third, pro-inflammatory cytokines have been reported to induce angiotensin type 1 receptors (AT1R) in cardiac fibroblasts, thus accentuating fibrogenic responses to neurohumoral stimuli [82]. Fourth, cytokines can induce the synthesis of matricellular proteins, such as tenascin-C [83], that serve as potent activators of fibroblast function [84]. Thus, it is not surprising that fibroblast-specific disruption of the type 1 IL-1 receptor (IL1R1), the only signaling receptor for IL-1, attenuated collagen deposition and reduced fibrosis in an infarction model [85]. 

#### 6.4.2. Growth Factors: The Role of TGF-βs, Fibroblast Growth Factors (FGF)s, and PDGFs

A large body of evidence supports the central role of TGF-β signaling cascades in reparative fibroblast activation and in myofibroblast conversion following myocardial infarction [101]. Although descriptive data from animal models of myocardial infarction demonstrate that all three TGF-β isoforms are markedly upregulated in the infarcted heart [102], their relative roles in the activation of infarct fibroblasts remains unknown. Both de novo synthesis of TGF-β isoforms and the activation of TGF-β from latent myocardial stores contribute to a marked increase in bioactive TGF-β following myocardial infarction. Matricellular proteins [103], proteases [104], and αv integrin-mediated actions [105] generate TGF-β bioactivity in pericellular areas, thus localizing the fibrogenic response. Subsequently, the active TGF-β dimer binds to a heterotetrameric complex comprised of type II and type I TGF-β receptors and activates downstream signaling pathways involving members of the Smad family of transcription factors or Smad-independent cascades [106]. In vitro studies demonstrated that Smad3 is critically involved in the activation of a matrix-synthetic cardiac fibroblast phenotype and in the conversion of fibroblasts into myofibroblasts [107]. However, in vivo, myofibroblast infiltration is not reduced in mice with global [107] or fibroblast-specific Smad3 loss [108]. In fact, Smad3 was found to mediate anti-proliferative actions that restrain the uncontrolled expansion of the myofibroblast population in the infarct [107,108]. The activating effects of Smad3 in infarct myofibroblasts were mediated predominantly by the activation of an integrin-dependent oxidative response that was essential for the formation of well-organized fibroblast arrays in the infarct, thus generating an ECM network comprised of aligned fibers [108]. These crucial effects of fibroblast-specific Smad3 signaling on the composition of the scar prevent cardiac rupture and attenuate adverse dilative remodeling [108]. In addition to actions on the mechanical properties of the ventricle, matrix-driven myofibroblast alignment may also play a role in signaling, transducing cascades of critical significance for the stimulation of matrix synthesis [109]. In contrast to the central role of Smad3 in the activation of reparative fibroblasts in the infarct, fibroblast-specific Smad2 signaling does not play a critical role in the repair and remodeling of the infarcted heart [110].

In addition to the effects of TGF-β superfamily members [111], several other growth factors have been implicated in the activation of infarct fibroblasts. Fibroblast growth factor (FGF)2 has been implicated in the proliferation of infarct fibroblasts in vitro and in vivo [112,113]. The platelet-derived growth factor system is also involved in fibroblast activation in the infarcted heart [114]. Antibody neutralization experiments suggested that PDGFRα signaling may promote fibroblast activation, whereas PDGFRβ may be important for maturation of the infarct vasculature, mediating the recruitment of mural cells by infarct neovessels [114]. In vitro, both PDGFRα and PDGFRβ activation were found to stimulate cardiac fibroblast proliferation and ECM protein synthesis [115,116]. 

#### 6.4.3. Neurohumoral Pathways

Both the systemic activation and local release of neurohumoral mediators play a prominent role in the regulation of fibroblast and myofibroblast phenotypes following myocardial infarction. The fibrogenic actions of the renin–angiotensin–aldosterone system (RAAS) have been extensively documented in infarcted hearts [117,118] and involve both direct actions and effects mediated through TGF-β [119,120]. Angiotensin II potently stimulates fibroblast proliferation and myofibroblast conversion and induces the synthesis of ECM proteins through the AT1R [121,122,123,124] and downstream activation of reactive oxygen species [125]. In contrast, the AT2 receptor has been suggested to exert anti-fibrotic actions, restraining AT1R-mediated fibroblast proliferation and ECM synthesis [126,127]. Despite the pronounced effects of angiotensin II on fibroblast activity, AT1R blockade in patients and in experimental models had no effect on the repair of the infarcted heart but exerted profound protective actions [128]. The molecular basis for the selectively maladaptive fibroblast responses triggered by AT1R remains unclear. Considering the broad effects of angiotensin II on cardiomyocytes, immune cells, vascular cells, and fibroblasts, it is not known to what extent attenuated fibrosis mediates the pro-survival effects of angiotensin-converting-enzyme inhibition or AT1R blockade in patients surviving acute myocardial infarction. 

Mineralocorticoid receptor activation by aldosterone is also implicated in fibroblast activation in the infarcted heart [129,130]. Aldosterone-mediated fibrogenesis may involve not only direct actions on cardiac fibroblast proliferation and ECM synthesis [131,132] but also effects on macrophages [133], T cells [134], or cardiomyocytes [135]. The protective effects of mineralocorticoid receptor inhibition in animal models and in patients with myocardial infarction are associated with attenuated fibrosis [136,137]. However, the relative involvement of primary anti-fibrotic actions in the protection afforded by mineralocorticoid receptor blockade remains unclear.

Although adrenergic stimulation primarily targets cardiomyocytes, some of its effects on the infarcted myocardium may involve actions on immune cells and fibroblasts, resulting in accentuated fibrosis. Activation of β2-adrenergic receptor signaling triggers cardiac fibroblast proliferation and activation [138,139,140]. However, to what extent the activation of infarct fibroblasts reflects adrenergic stimulation is unknown. 

#### 6.4.4. Matricellular Proteins in Activation of Infarct Fibroblasts: The ECM as a Signaling Hub

In the healing infarct, the ECM network not only serves a structural role but also transduces key signals that regulate cell activation [141]. This transformation of the matrix into a signaling hub involves the induction and deposition of matricellular proteins, a family of structurally unrelated proteins that do not play a primary structural role but are induced following injury and modulate signaling responses [141]. Several matricellular proteins and other specialized ECM macromolecules, including osteopontin, tenascin-C, thrombospondin-1, osteoglycin, periostin, SPARC (secreted protein acidic and rich in cysteine), cartilage intermediate layer protein 1 (CILP)1, and ED-A fibronectin, enrich the healing infarct and transduce important activating signals in cardiac fibroblasts [84,142,143,144,145,146,147,148,149], stimulating myofibroblast conversion and promoting collagen synthesis. The effects of these multidomain proteins are mediated through several different mechanisms. First, matricellular proteins bind to latent growth factors, such as TGF-β, contributing to the release of the active molecule [150] and localizing TGF-β bioactivity in the area of injury. Second, several members of the matricellular family bind to integrins or other receptors (such as CD36 or CD44) on the fibroblast surface, transducing activating signals [151]. Third, some matricellular proteins (such as thrombospondin-1) bind to proteases, regulating their activity [152]. 

#### 6.4.5. The Intracellular Network of Molecular Cascades Involved in Fibroblast Activation

Growth factors, neurohumoral mediators, mechanical stress, and matricellular signals initiate cascades (Table 2) that ultimately converge on the activation of several intracellular cascades with a central role in the regulation of the fibroblast phenotype [153]. In addition to the Smad cascades (which were discussed in the section on the role of TGF-β), mitogen-activated protein kinases (MAPKs), Rho kinase, and the Yes-associated protein 1 (YAP)/Transcriptional coactivator with a PDZ-binding motif (TAZ) system have been extensively implicated in the activation of fibroblasts under conditions of stress. 

The central involvement of MAPKs in infarct fibroblast activation is supported by both in vitro and in vivo studies. p38α MAPK is the predominant isoform expressed in cardiac fibroblasts [154] and was found to promote myofibroblast conversion in vitro and in vivo following myocardial infarction. p38 MAPK-mediated fibrogenic actions involve the activation of the myocardin-related transcription factor (MRTF)/serum response factor (SRF) axis [140,155,156,157]. 

The activation of the small GTP-binding protein RhoA and the subsequent stimulation of the Rho-associated coiled-coil containing kinases (ROCKs), ROCK1 and ROCK2, are important regulators of the fibroblast phenotype that link integrin activation and cytoskeletal dynamics. The RhoA/ROCK system may play an important role in a wide range of fibroblast responses that are involved in reparative fibrosis, including proliferation, differentiation, and migration [158]. However, most of the evidence implicating fibroblast-specific RhoA-ROCK in cardiac fibrotic conditions is derived from investigations in models of left ventricular pressure overload or angiotensin infusion [159,160]. The potential involvement of RhoA signaling in the activation of infarct myofibroblasts has not been studied. 

The mechanosensitive YAP–TAZ pathway is regulated both by the Hippo pathway and by Hippo-independent mechanisms and has recently been implicated in the activation of the fibroblast phenotype in infarcted and remodeling hearts. Fibroblast-specific loss-of-function approaches documented that YAP–TAZ signaling activates infarct fibroblasts, acting downstream of TGF-β cascades [161] through the engagement of TEA domain transcription factor 1 and the subsequent de novo expression of MRTF A [162]. The pro-fibrotic actions of YAP–TAZ were found to be maladaptive, resulting in adverse remodeling and dysfunction without a central involvement in the reparative response [161,162].

**Table 2 cells-11-01386-t002:** Studies documenting fibroblast-mediated signaling in repair and remodeling of the infarcted heart using fibroblast-specific targeting approaches.

Mediator	Fibroblast-Specific Approach	Role in Repair and Remodeling of the Infarcted Heart	Role in Modulation of Fibroblast Phenotype	Proposed Mechanism	Ref.
IL1-R1	Tamoxifen-inducible Col1a2 CreERT mice were used for fibroblast- specific deletion of IL1-R1.	Fibroblast-specific IL1-R1 drives adverse cardiac remodeling and promotes ventricular wall thinning and collagen deposition in a model of non-reperfused infarction.	IL1-R1 promotes a pro-fibrotic and matrix-degrading phenotype in fibroblasts.	IL1-α interacts with IL1-R1 to activate p38 and NFκB signaling, leading to increased expression of IL6, MMP3, and MMP9 and secretion of IL-6 and MMP-3.	[85]
Smad3	Myofibroblast-specific deletion used Postn-Cre mice.	Myofibroblast-specific Smad3 protects from cardiac rupture in the non-reperfused infarction model and attenuates chamber dilation.	Smad3 limits fibroblast proliferation and promotes collagen synthesis. It also mediates the formation of organized myofibroblast arrays.	Smad3 regulates fibroblast function via integrin-mediated NOX-2 expression. Moreover, Smad3-dependent activation of the GTPase RhoA dictates fibroblast alignment via the regulation of cell polarity pathways.	[108,110]
Smad7	Postn-Cre mice were used for myofibroblast-specific knockout of Smad7.	Myofibroblast Smad7 protects the infarcted heart from adverse remodeling and from heart failure-related death. Smad7 limits post-infarction fibrosis in the border zone and in the papillary muscles.	Smad7 attenuates myofibroblast activation and the synthesis of structural and matricellular ECM proteins.	Smad7 inhibits the TGFβ response via the inactivation of Smad2/3. Smad7 also binds to ErbB2 and restrains the activation of ErbB1/2 in a TGF-β-independent manner to suppress the expression of fibrogenic genes.	[5]
GSK3β	Postn-Cre mice and tamoxifen-inducible Col1a2-Cre mice were used for deletion in myofibroblasts and fibroblasts, respectively.	Myofibroblast-specific GSK3β negatively regulates fibrosis to limit adverse ventricular remodeling in the infarcted heart.	GSK3β functions to suppress myofibroblast activation and pro-fibrotic signaling.	GSK3β inhibits TGF-β-dependent Smad3 transcriptional activation to limit fibrogenic signaling.	[163]
P38α	TCF-21 or Postn-MCM mouse models were used to generate fibroblast and myofibroblast-specific p38α KO mice.	P38α-dependent signaling in fibroblasts drives fibrosis to promote adverse post-infarction remodeling.	P38 mediates cardiac fibroblast activation.	P38 transduces mechanical and cytokine signals via serum response factor and calcineurin to promote myofibroblast differentiation.	[140]
GRK2	Myofibroblast-specific GRK2 deletion was achieved using Postn-MCM mice.	GRK2 signaling contributes to pathological cardiac remodeling via promoting fibrosis and infarct expansion, leading to cardiac dysfunction.	GRK2 promotes myofibroblast activation and collagen deposition in vivo.	GRK2 interacts with Gβγ, promoting the downregulation of fibroblast β-adrenergic receptors. This decreases downstream cAMP production, resulting in the activation of pro-fibrotic signals.	[164]
Lats1/2	Tcf21-MCM mice were used for fibroblast-specific Lats1/2 deletion.	Lats1/2 limit pro-fibrotic signaling.	Lats1/2 maintain the resting fibroblast phenotype and prevents activation to myofibroblast.	Lats1/2 act to maintain the quiescent fibroblast cell state via inhibiting the YAP-induced activation of pro-fibrogenic genes.	[31]
YAP	Tcf21-MCM and Col1a1 CreERT mice were used for YAP deletion in fibroblasts.	YAP promotes post-infarction fibrosis.	YAP activation in fibroblasts promotes myofibroblast proliferation and differentiation and ECM gene expression.	YAP activation induces MRTF-A expression to facilitate myofibroblast differentiation and profibrotic gene expression.	[162]
5-HT2B	Tcf21-MCM and Postn-MCM were used to delete 5-HT2B in resting and activated fibroblasts, respectively.	5-HT2B expression directly contributes to excessive scar formation, leading to adverse cardiac remodeling and impaired cardiac function post-MI.	5-HT2B promotes fibroblast proliferation and migration and the expression of ECM remodeling genes.	5-HT2B-dependent fibroblast responses are mediated via Dnajb4 expression and Src phosphorylation.	[165]
Hsp47	Postn-MCM mice were used for the generation of myofibroblast-specific Hsp47 KO mice.	Hsp47 expression in myofibroblasts mediates scar formation post-MI.	HSP47 promotes fibroblast proliferation and mediates the expression of collagens without affecting the expression of α-SMA.	HSP47 enhances the expression of Snail and Zeb1 to transcriptionally activate ECM-related genes. It also downregulates the expression of cell cycle inhibitory kinases to facilitate cell proliferation.	[32]
Sox9	Postn-MCM mice were used for myofibroblast-specific KO.	Myofibroblast-specific Sox9 facilitates MI-induced left ventricular dysfunction, inflammation, and tissue scarring.	Sox9 activity promotes fibroblast activation, proliferation, migration, and contractile function.	Sox9 up-regulates ECM-related gene synthesis, inflammation, and proteolysis.	[166]
AMPKα1	Postn-Cre was used for the deletion of AMPKα1 in myofibroblasts.	AMPKα1 activation in myofibroblasts limits adverse post-infarction remodeling post-MI.	Myofibroblast-specific AMPKα1 inhibits fibroblast proliferation in response to injury.	AMPKα1 inhibits fibroblast activation and proliferation via the miR-125b-5p-dependent silencing of connexin-43.	[167]
Muscleblind-like1 (MBNL1)	Tcf21-MCM was used for the fibroblast-specific deletion and overexpression of MBNL1. Postn-MCM was used for the deletion of MBNL1 in activated fibroblasts.	Myofibroblast-specific MBNL1 facilitates the acute wound-healing response post-MI and promotes tissue fibrosis.	MBNL1 promotes myofibroblast transition and contractile function in fibroblasts.	The RNA-binding protein MBNL1 binds to and stabilizes mRNA encoding CnAβ and SRF, promoting myofibroblast differentiation and profibrotic gene expression.	[168]
Fibronectin	Tcf21-MCM was used for the ablation of fibronectin in fibroblasts.	Fibronectin polymerization facilitates adverse cardiac remodeling and fibrosis post I/R injury.	Polymerized FN promotes fibroblast proliferation and migration and collagen matrix deposition.	Fibronectin activates c-myc signaling, leading to integrin β1 activation and the downstream phosphorylation of FAK.	[169]

### 6.5. Angiogenic Effects of Infarct Fibroblasts

Neovessel formation is important in healing the infarcted heart [170], as it supplies the metabolically active granulation tissue cells with the oxygen and nutrients necessary for their reparative functions. Several lines of evidence suggest that infarct fibroblasts can secrete angiogenic mediators, contributing to neovessel formation. Fibroblasts harvested from infarcted hearts 3 days after coronary occlusion secreted angiogenic mediators, such as vascular endothelial growth factor (VEGF)-A, whereas, at later timepoints, fibroblasts acquired an angiostatic profile, expressing large amounts of the angiogenesis inhibitor thrombospondin-1 [171]. Moreover, a subpopulation of FSP1+/CD31-/CD45- fibroblast-like cells has been identified in the infarcted myocardium and was found to express potent angiogenic mediators, such as VEGF and stromal cell-derived factor (SDF)1/CXCL12 [172]. These angiogenic fibroblasts are distinct from the matrix-secreting myofibroblasts, suggesting a functional specialization of fibroblast subsets in the infarcted heart.

In addition to effects mediated through the secretion of angiogenic growth factors, fibroblasts have also been suggested to directly contribute to angiogenesis by converting to endothelial cells. Lineage-tracing experiments in healing infarcts suggested that fibroblasts may transdifferentiate to endothelial cells through the activation of a p53-mediated pathway [173], thus serving a direct role in angiogenesis. The notion that fibroblasts are highly plastic cells that can acquire a vascular cell phenotype is also supported by experiments identifying fibroblast-like cells with endothelial cell characteristics in the ischemic limb [174] and by studies showing endothelial lineage conversion in human fibroblasts, mediated through the expression of the transcription factor SOX17 [175]. However, this concept remains controversial, as a very systematic in vivo study, using several different inducible Cre drivers to label fibroblasts and endothelial cells, demonstrated no significant contribution of fibroblast-to-endothelial-cell conversion in the neovascularization of the healing infarct [176]. Pre-existing endothelial cells were identified as the main source of neovascular endothelium in infarcted hearts [176].

## 7. Fibroblasts during the Maturation Phase of Infarct Healing

### Matrifibrocyte Transition and Negative Regulation of the Fibrotic Response

The deposition of structural collagens in the infarct is associated with the induction of matrix cross-linking enzymes that contribute to the formation of a mature scar. At this stage, myofibroblasts become quiescent and ultimately disassemble the α-SMA+ stress fibers [36]. Although a fraction of infarct myofibroblasts may undergo apoptosis [177], the majority of these cells convert to matrifibrocytes (Figure 3), specialized fibroblast-like cells that express high levels of cartilage and tendon ECM proteins [38]. Although the signals mediating matrifibrocytes transition are unknown, several molecular mediators have been implicated in the endogenous suppression of the fibrogenic profile of infarct myofibroblasts, thus serving as negative regulators of the fibrotic response

Considering the central role of TGF-β/Smad signaling cascades in the fibrotic response, it is not surprising that signals inhibiting TGF-β-driven myofibroblast activation play an important role in restraining post-infarction fibrosis. The inhibitory Smads, Smad6 and Smad7, are important inducible negative regulators of TGF-β signaling cascades, acting as competitive inhibitors of receptor-activated Smad activation and nuclear translocation through binding to type I TGFβ receptors [178] or Smad4, [179] or by mediating type 1 TGF-β receptor degradation by recruiting the ubiquitin ligases Smurf 1 and 2 [180]. Smad7 expression is specifically upregulated in α-SMA+ myofibroblasts infiltrating the infarcted heart, with much lower expression in α-SMA-negative/PDGFRα+ fibroblasts [5]. Studies using cell-specific knockout mice demonstrated that myofibroblast Smad7 expression protects the infarcted myocardium from heart failure and attenuates adverse remodeling and fibrosis by restraining collagen accumulation and myofibroblast conversion. Surprisingly, the anti-fibrotic effects of Smad7 were only partially related to the inhibition of TGF-β cascades but also involved TGF-β-independent inhibitory actions of Smad7 on the receptor tyrosine kinase Erbb2 [5]. Whether Smad6 is also involved in the negative regulation of the fibroblast phenotype in the healing infarct is unknown.

In addition to the effects of the inhibitory Smads, several other anti-fibrotic pathways have been identified as important negative regulators of post-infarction fibrosis. The nuclear TGF-β repressor c-Ski is markedly upregulated in the infarcted myocardium during the maturation phase of infarct healing [181,182]. The intracellular kinase AMP-activated protein kinase (AMPK)a1 was found to restrain fibroblast activation and proliferation following myocardial infarction through effects that were attributed to connexin 43 upregulation [167]. Components of the ECM may also contribute to the deactivation of fibroblasts in the infarct. Collagen V was found to reduce scar expansion after infarction by restraining the expression of mechanosensitive integrins in infarct fibroblasts [183]. Considering the significance of the negative regulation of the fibroblast phenotype in the infarcted heart, several distinct inhibitory pathways may co-operate in the infarcted heart to induce the quiescence of infarct myofibroblasts while retaining a fibroblast population responsible for scar maintenance.

## 8. Fibroblast Heterogeneity in Infarcted Hearts

In addition to their dynamic transitions in response to changes in their microenvironment, fibroblasts also exhibit remarkable heterogeneity. Over the last 5 years, single-cell transcriptomic studies have contributed important new information on the diversity of fibroblasts in normal and infarcted hearts [184] by defining clusters with distinct transcriptional profiles [185]. A population of fibroblasts expressing high levels of *Cthrc1* (the gene encoding collagen triple helix repeat containing 1, a secreted glycoprotein that promotes cell migration [186]) emerged after infarction and was implicated in repair and protection from rupture [187]. Moreover, in a model of neurohumoral activation, two distinct clusters expressing *Cilp* and *Thbs4,* respectively, emerged as potentially important pro-fibrotic populations in the absence of myofibroblast conversion [188]. Although single-cell transcriptomics have revolutionized our perspective on interstitial cells, major challenges remain in understanding the relations between transcriptional profiles and the functional properties of the various cell subpopulations.

## 9. Chronic Activation of Fibroblasts in the Remodeling of Non-Infarcted Myocardium

During infarct healing, the surviving remote myocardium remodels, exhibiting compensatory hypertrophic changes accompanied by macrophage activation and interstitial fibrotic changes. Extensive descriptive data in both rodent and large animal models of myocardial infarction suggest that fibroblast activation is much more prominent in border zone areas adjacent to the healing infarct, with lower levels of activation in remote remodeling myocardial segments [189,190,191,192]. Moreover, the infarct border zone exhibits evidence of collagen denaturation, which may reflect mechanical stress in the interface between viable myocardium and non-contractile scar [53]. The extent of fibrotic interstitial remodeling in non-infarcted segments is dependent on the size of the infarct and on the severity of the hemodynamic perturbations.

The cellular mechanisms responsible for fibroblast activation in the remote remodeling myocardium remain poorly understood, as experimental interventions targeting the fibroblasts have a major impact on infarct healing, precluding conclusions regarding specific events occurring in fibroblasts populating the non-infarcted myocardium. Fibroblast activation in remote remodeling myocardium may reflect the increased wall stress in non-infarcted myocardial segments and may contribute to progressive dysfunction and the development of heart failure after a large myocardial infarction.

## 10. Fibroblasts and Myofibroblasts as Therapeutic Targets in Myocardial Infarction

Extensive experimental evidence suggests a dual role for activated fibroblasts in the infarcted myocardium. During the proliferative phase, activated myofibroblasts play a crucial reparative role, preventing cardiac rupture and attenuating adverse dilative remodeling by forming an organized matrix network, which increases the tensile strength of the infarct. On the other hand, excessive, unrestrained or expanded fibroblast activation stimulates progressive fibrotic remodeling in the infarct border zone and the remote remodeling myocardium, contributing to dysfunction and to the pathogenesis of chronic heart failure. The dual role of fibroblasts in the infarcted heart poses a major challenge for the therapeutic implementation of strategies targeting fibrosis following myocardial infarction. Clearly, early inhibition of fibroblast activation during the proliferative phase of cardiac repair is likely to have detrimental effects by abrogating key reparative responses. This concern may be more important for elderly patients who may exhibit age-associated perturbations in the activation of reparative fibroblasts [193]. Late inhibition of fibrotic remodeling may be effective in attenuating heart failure progression in patients exhibiting prominent fibroblast activation. Imaging studies and biomarkers reflecting collagen synthesis and remodeling may represent important tools for the identification of patients that may benefit from anti-fibrotic interventions [194].

Over the last 10 years several studies attempted to regenerate the infarcted myocardium by reprogramming fibroblasts into functional cardiomyocyte-like cells [195,196,197] using a variety of approaches that combined transcription factors, microRNAs, and small molecules. Although this direction holds promise, major advances in understanding the fundamental biology of reprogramming are needed [198] in order to address challenges related to the low efficiency of the approaches and the phenotypic and functional characteristics of the reprogrammed cells.

## 11. Conclusions

Fibroblasts are critical effector cells in the repair of the infarcted heart but are also implicated in chronic fibrotic remodeling after myocardial infarction. A growing body of evidence suggests that the heterogeneity of fibroblast populations may account, at least in part, for their diverse functional profiles. Targeting fibroblasts in myocardial infarction requires an understanding of the relative roles of the specific subsets in the regulation of the reparative and maladaptive responses. Moreover, in the clinical context, we need to identify patient subpopulations with perturbations of reparative fibroblasts and subjects with progressive fibrotic remodeling who may benefit from interventions targeting the fibroblasts. 

## Figures and Tables

**Figure 1 cells-11-01386-f001:**
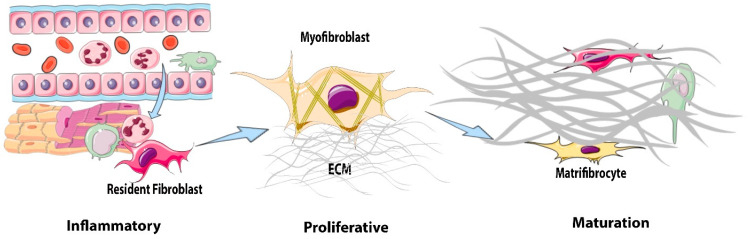
**Phenotypic transitions of cardiac fibroblasts in the infarcted myocardium.** During the inflammatory phase of infarct healing, necrotic cardiomyocytes release damage-associated molecular patterns that activate a pro-inflammatory, matrix-degrading phenotype in cardiac fibroblasts and contribute to the recruitment of leukocytes to the injury site. The clearance of dead cardiomyocytes stimulates anti-inflammatory signals, leading to the transition to the proliferative phase of infarct healing. During the proliferative phase of infarct healing, resident cardiac fibroblasts proliferate and undergo myofibroblast conversion, incorporating α-smooth muscle actin (α-SMA) into cytoskeletal stress fibers. Myofibroblasts are the main matrix-synthetic cells in the infarcted heart and produce both structural and matricellular extracellular matrix (ECM) proteins. During scar maturation, fibroblasts exhibit the disassembly of stress fibers and transition to matrifibrocytes, specialized cells that may play a role in scar maintenance.

**Figure 2 cells-11-01386-f002:**
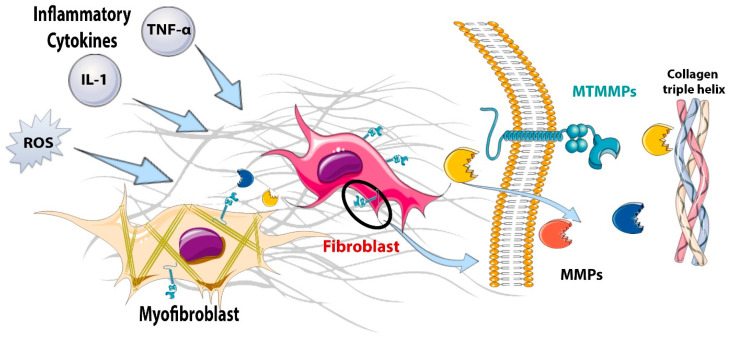
**The matrix-degrading properties of fibroblasts in the infarcted myocardium.** During the inflammatory phase of infarct healing, pro-inflammatory cytokines, such as interleukin (IL)-1β, tumor necrosis factor (TNF)-α, and IL-6 stimulate the expression and secretion of matrix metalloproteinases (MMPs) that degrade the extracellular matrix, setting the stage for the replacement of damaged tissue with a collagen-based scar. During the inflammatory and early proliferative phases of infarct healing, the induction of membrane-type matrix metalloproteinases (MT-MMPs) on the cell surface plays an important role in fibroblast migration, thus localizing the reparative fibroblasts to the area of infarction.

**Figure 3 cells-11-01386-f003:**
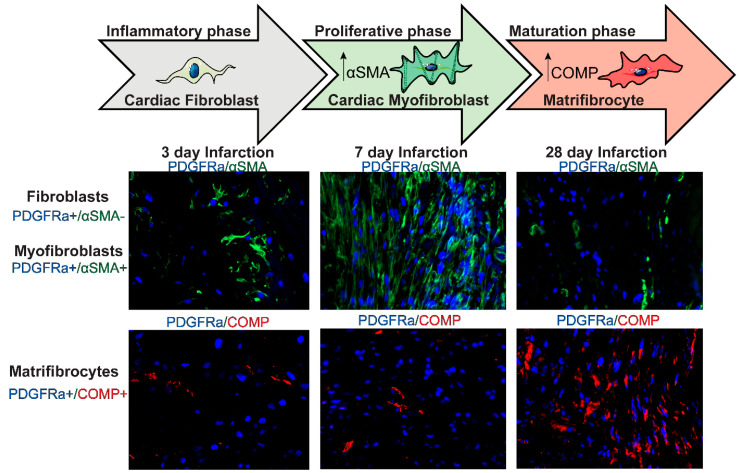
**Dynamics of fibroblast activation during the proliferative and maturation phases of infarct healing.** During the inflammatory phase of infarct healing, resident cardiac fibroblasts (identified as PDGFRα+ cells in PDGFRα^EGFP^ reporter mice) do not express the myofibroblast marker a-smooth muscle actin (α-SMA). During the proliferative phase (3–14 days), many fibroblasts become activated, proliferate, and differentiate into cardiac myofibroblasts (PDGFRα+/α-SMA+) that secrete large amounts of extracellular matrix proteins. During the maturation phase of infarct healing (14 days–2 months), myofibroblasts disassemble the α-SMA+ stress fibers and convert to matrifibrocytes, specialized cells expressing bone-cartilage proteins, such as cartilage oligomeric matrix protein (COMP).

**Table 1 cells-11-01386-t001:** Effects of pro-inflammatory cytokines on cardiac fibroblasts.

Cytokine/Species	Effect	Mechanism	Ref.
IL-6/rat cardiac fibroblasts	IL-6 enhances collagen synthesis and myofibroblast formation.	IL-6 binds to gp130, leading to the phosphorylation of Janus kinase, activating cellular events.	[86]
IL-6/mouse cardiac fibroblasts	IL-6 promotes myofibroblast differentiation and the release of proinflammatory cytokines in culture. In vivo, treatment with an anti-IL-6 blocking antibody reduced myofibroblast infiltration in the infarct (and also attenuated neutrophil infiltration) but also increased the size of the infarct.	IL-6-induced phosphorylation of STAT3 upregulates the expression of hyaluronan synthesis that supports a myofibroblast phenotype in cultured fibroblasts.	[87]
IL-6/mouse cardiac fibroblasts	IL-6 promotes fibroblast activation and collagen synthesis.	In a co-culture model, macrophages stimulate cardiac fibroblasts to produce IL-6, which promotes TGF-β production and the downstream activation of Smad3 in fibroblasts.	[80]
IL-6/neonatal rat ventricular fibroblasts	IL-6 stimulates fibroblast proliferation and myofibroblast differentiation under hypoxia. Inhibition of IL-6 signaling with an IL-6 receptor inhibitor attenuates hypoxia-induced fibroblast proliferation and differentiation and collagen I expression.	In cultured fibroblasts exposed to hypoxia, the effects of IL-6 are attributed to the activation of TGFβ1, MMP2, and MMP9.	[88]
IL-6/human cardiac fibroblasts	IL-6 in endothelial cell-derived conditioned media increases collagen type I and fibronectin gene expression in cardiac fibroblasts. The addition of soluble gp130 to endothelial cell-derived conditioned media prevents IL-6-dependent collagen type I and fibronectin gene expression.	IL-6 in conditioned media from endothelial cells binds to soluble IL-6R to induce trans-IL-6 signaling in cardiac fibroblasts.	[89]
IL-6/rat cardiac fibroblasts	In contrast to the promigratory effects of IL-1β and TNF-α, IL-6 has no effect on fibroblast migration.	IL-6 does not stimulate the activation of mitogen-activated protein kinases that are involved in the regulation of cell migration.	[40]
IL-6/adult mouse ventricular fibroblasts.	IL-6 increases fibroblast adhesion and proliferation.	Cardiomyocyte-derived IL-6 acts in a paracrine manner to promote fibroblast proliferation in a cardiomyocyte/fibroblast co-culture model.	[90]
IL-6/neonatal mouse cardiac fibroblasts	IL-6 loss decreases fibroblast-myocyte adhesion in vitro and markedly upregulates fibroblast proliferation.	In a fibroblast/cardiomyocyte coculture system, IL-6/soluble IL-6R trans-signaling activates STAT3 in fibroblasts to modulate fibroblast proliferation and adhesion to cardiomyocytes.	[91]
IL1-β/mouse cardiac fibroblasts	IL1-β: (a) attenuates TGF-β-induced α-SMA expression and incorporation into stress fibers, (b) abrogates fibroblast-mediated collagen pad contraction and expression of periostin, and (c) promotes a matrix-degrading phenotype via stimulating MMP3 and MMP8 expression.	IL1-β acts via IL-1R1 to upregulate BAMBI, which acts to negatively regulate TGFβ signaling. IL1-β also downregulates endoglin signaling receptors.	[35]
IL1-β/adult rat cardiac fibroblasts	IL1-β augments the expression and activity of MMP-2, 3, and 9, alongside an increase in TIMP-1 expression, and enhances fibroblast migration.	IL1-β activates p38, ERK, and JNK MAP kinase pathways to stimulate MMP expression and migration.	[92]
IL-1β/neonatal and adult rat cardiac fibroblasts	IL1-β selectively downregulates the expression and synthesis of fibrillar collagens. Increases total MMP activity, with an increase in the expression of MMP-2, 9, and 13.	No mechanism is studied.	[93]
IL-1β/mouse cardiac fibroblasts	IL-1β stimulates proinflammatory gene expression. It promotes ECM remodeling.	IL-1β acts via IL-R1 to promote ECM remodeling via enhancing the fibroblast expression of MMPs (MMP-3, 8, and 9) and downregulating the expression of TIMP-2 and TIMP-4.	[94]
IL-1β/neonatal rat cardiac fibroblasts	IL-1β induces AT1 receptor upregulation in fibroblasts, contributing to ECM remodeling.	IL-1β acts via an NFκB-dependent mechanism to upregulate AT1R expression.	[95]
IL-1α/mouse neonatal ventricular fibroblasts	IL-1R antagonism and the administration of an anti-IL-1α blocking antibody show that the conditioned medium of necrotic cardiomyocytes activates proinflammatory signaling in fibroblasts through IL-1α.	IL-1α acts via an MyD88-dependent and NLRP3-independent pathway to promote pro-inflammatory gene expression in cardiac fibroblasts.	[43]
IL-1α/human cardiac fibroblasts	IL-1α markedly increases the expression of MMP-1, 3, 9, and 10, with a minimal effect on the mRNA expression of structural ECM proteins, and reduces the expression of ADAMTS1.	IL-1α acts via distinct P38 MAPK subtypes α/β/γ/δ to regulate the expression of MMPs and metalloproteinases in fibroblasts.	[96]
IL-1α/human cardiac fibroblasts	IL-1α stimulates proinflammatory gene expression in fibroblasts via upregulation of IL-1β, TNF-α, and IL-6.	ERK, JNK, and p38 MAPKs, along with nuclear factor (NF)-kB signaling, distinctly regulates IL1-β, TNF-α, and IL-6 expression.	[39]
IL-1α/human cardiac fibroblasts	Cardiac fibroblasts express neutrophil-binding adhesion molecules and neutrophil chemoattractants in response to IL-1α, promoting neutrophil recruitment into the infarcted myocardium.	IL-1α acts via a p38- and NF-κB-dependent mechanism to promote the expression of ICAM-1, E-selectin, and CXC chemokines in fibroblasts.	[97]
IL-1α/human cardiac fibroblasts	IL-1α has opposing effects on the expression of connective tissue growth factor (CTGF) and tenascin-C (TNC) expression.	Stimulates NFκB, PI3K/AKT, and p38 MAPK pathways to upregulate the expression of TNC while downregulating CTGF expression.	[83]
TNF-α/neonatal and adult rat cardiac fibroblasts	TNF-α promotes matrix degradation via mediating a decrease in collagen synthesis with an increase in MMP-2, MMP-9, and MMP-13. It has no effect on cell proliferation and total protein synthesis.	No mechanism studied.	[93]
TNF-α/neonatal rat cardiac fibroblasts	TNF-α increases AT1 receptor density in cardiac fibroblasts.	TNF-α acts via NFκB to increase AT1 receptor expression.	[95]
TNF-α/primary human fibroblasts from patient biopsies with dilated cardiomyopathy	(a) TNF-α increases cytokine expression at the transcriptome level; however, this increase was not reflected in the cytokine secretome. (b) TNF-α treatment has no effect on collagen/MMP/TIMP gene expression.	TNF-α effects are mediated via the transcriptional activation of NFκB.	[98]
TNF-α/rat cardiac fibroblasts	TNF-α stimulates a concentration-dependent increase in fibroblast migration.	TNF-α-dependent migration is regulated by the activation of MAPKs–ERK1/2, JNK, and p38.	[40]
TNF-α/human cardiac fibroblasts	TNF-α promotes fibroblast MMP-9 expression that is abrogated following treatment with an anti-TNF-α blocking antibody.	TNF-α acts via NFκB to promote the expression of MMP-9.	[99]
TNF-α/human cardiac fibroblasts	TNF-α promotes the proliferation of cardiac fibroblasts.	TNF-α-dependent activation of ERK1/2 and NFκB drives fibroblast cell proliferation.	[100]

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
