# Peer review of "Properties and Functions of Fibroblasts and Myofibroblasts in Myocardial Infarction"

_cells, 2022, doi:10.3390/cells11091386_

Round 1
Reviewer 1 Report
The review article “Fibroblasts and myofibroblasts in myocardial infarction” is well written and easy to read review which describes important role of fibroblasts in post-infarct heart as a crucial part of cell cohort. Review based on analysis of comprehensive number of papers and overlooked wide range of mechanistic pathways in reparative process in post-infarct condition. It may was just typos but in some case such as in line 332 and 339 looked like some missing part of words.
Regardless comments above I would like to say that this one of good review which is refreshing knowledge in the field for established scientists as much as helpful information for young researchers because points on unknown and not-understood aspects of biology of fibroblast. These grey patches in the filed can be good source of new exciting project.
I wish authors success and recommend publishing paper in such reputable journal.
Reviewer 2 Report
- Although the paper is well written the topic on the myocardial infarction is interesting, the manuscript has to be improved. The title should be more specific because the central theme is myocardial infarction. It should possibly refer to the heterogeneity of fibroblasts / myocytes during infarction.
- The references must be updated (mostly within the last 5 years) and arranged in order.
- Line 282: Please fill in the name of the TNF factor
- Add to the table a column with examples of anti cytokines or enzyme inhibitors that interfere with myocardial cellular events.
